# Rapeseed Protein Isolate as a Sustainable Alternative to Soy Protein: A Case Study on Chicken Pâtés

**DOI:** 10.3390/foods14223841

**Published:** 2025-11-10

**Authors:** Predrag Ikonić, Miloš Županjac, Branislava Đermanović, Miroslav Hadnađev, Mladenka Pestorić, Ljubiša Šarić, Nikola Maravić, Bojana Šarić

**Affiliations:** Institute of Food Technology in Novi Sad, University of Novi Sad, Bulevar cara Lazara 1, 21000 Novi Sad, Serbia; predrag.ikonic@fins.uns.ac.rs (P.I.); milos.zupanjac@fins.uns.ac.rs (M.Ž.); branislava.djermanovic@fins.uns.ac.rs (B.Đ.); miroslav.hadnadjev@fins.uns.ac.rs (M.H.); mladenka.pestoric@fins.uns.ac.rs (M.P.); ljubisa.saric@fins.uns.ac.rs (L.Š.); nikola.maravic@fins.uns.ac.rs (N.M.)

**Keywords:** chicken pâté, rapeseed protein isolate, soy protein isolate, storage stability, texture, fatty acids, rheology, sensory evaluation

## Abstract

This study investigated the potential of rapeseed protein isolate (RPI) as a sustainable alternative to soy protein isolate (SPI) in chicken pâtés, considering the combined effects of protein (SPI or RPI) and oil (sunflower or rapeseed) sources. The proximate composition, pH, water activity (a_w_), and colour of the RPI-based formulations were largely comparable to their SPI counterparts and remained stable over 60 days of refrigerated storage. However, at day 0, the RPI-based samples exhibited lower pH values (approx. 6.09 vs. 6.41), slightly lower lightness (*L** approx. 68.9 vs. 72.5), higher redness (*a** approx. 4.72 vs. 3.58), and higher yellowness (*b** approx. 23.3 vs. 9.38), indicating some initial formulation-dependent differences. Furthermore, RPI-based formulations exhibited enhanced textural properties and improved oxidative stability, likely due to synergistic effects between rapeseed protein and oil. The substitution of sunflower oil with rapeseed oil significantly improved the fatty acid profile (*p* < 0.05), notably increasing α-linolenic acid (C18:3n-3) and achieving a favourable n-6/n-3 ratio (approx. 2.8). Sensory evaluation revealed that the formulation combining both rapeseed ingredients provided a stable and highly acceptable profile. These results collectively demonstrate that RPI is a viable and functional replacer for SPI, enabling the production of nutritious, high-quality, and sustainable chicken pâtés.

## 1. Introduction

The incorporation of plant-based protein isolates into meat products has gained increasing attention due to growing consumer interest in health, sustainability, and ethical aspects of food production. Protein isolates derived from soy, pea, wheat, and various legumes exhibit key functional properties such as water-binding, emulsification, and gelation, which play a crucial role in maintaining the texture and sensory attributes of meat and meat analogues [1,2,3]. Beyond their functional role, plant proteins offer nutritional advantages, providing essential amino acids while being naturally free of cholesterol and lower in saturated fats compared with animal-derived proteins. As the meat industry strives to satisfy consumer demand for healthier and more sustainable options, the strategic integration of plant-based protein isolates presents both opportunities and challenges in food product development [1].

Soy protein isolate (SPI) is among the most widely used plant-based proteins in meat and meat analogue products due to its excellent functional and nutritional properties. With a protein content typically exceeding 90% and a balanced amino acid profile, SPI also exhibits strong water- and fat-binding, emulsifying, and gelling capacities, making it particularly effective in enhancing the texture, yield, and stability of processed meat systems [2]. From a nutritional perspective, SPI is considered a complete protein source, rich in essential amino acids such as lysine and arginine, and is associated with potential health benefits, including cholesterol-lowering effects and improved cardiovascular health [4]. Nevertheless, despite these advantages, SPI also presents several limitations that have driven interest in alternative protein sources. Key concerns include allergenicity, as soy is a common food allergen, potential off-flavours, and the presence of anti-nutritional factors such as trypsin inhibitors and phytic acid [5,6]. Growing consumer demand for soy-free, non-GMO, and sustainable protein sources has led to increased interest in alternative plant proteins, such as pea, faba bean, and lupin, for use in meat products. This shift is driven not only by health and sustainability considerations but also by regulatory developments, notably the EU Deforestation Regulation, which has placed additional scrutiny on soy cultivation and sourcing [1,7]. As a result, food manufacturers are exploring a diverse range of plant proteins to maintain product quality, meet consumer expectations, and ensure compliance with emerging environmental standards.

Within this context, rapeseed (*Brassica napus* L.), the second most widely cultivated oilseed crop worldwide [8], has gained recognition as a promising source of plant-based proteins [3,9,10,11]. This interest is largely attributed to rapeseed’s favourable amino acid composition and the substantial protein content found in its by-products, such as press cake and meal [12]. Protein isolates and concentrates derived from rapeseed have demonstrated considerable potential as meat extenders, contributing to improved nutritional value, desirable techno-functional characteristics, and greater sustainability in meat products [13,14,15,16,17,18]. In recent years, a growing number of studies have focused on the recovery and functional characterization of rapeseed protein isolates (RPI), highlighting their emulsifying, foaming, and water-binding properties comparable to those of soy and pea proteins. Moreover, advances in extraction and purification techniques have significantly improved RPI solubility, digestibility, and sensory profile, further reinforcing their industrial relevance [19,20]. The dominant protein fractions in rapeseed are the storage proteins cruciferin and napin, which together represent around 85–90% of the total protein content, alongside minor amounts of structural and metabolic proteins. Cruciferin is a high-molecular-weight 11S globulin (≈300–350 kDa) that largely determines the emulsifying and foaming behavior of rapeseed protein isolates, while napin, a low-molecular-weight 2S albumin (≈14–16 kDa), contributes to solubility and gelation. Owing to this complementary functionality, rapeseed proteins can provide desirable techno-functional properties across a wide range of applications, from emulsified and aerated systems to structured food matrices. These characteristics further highlight the potential of rapeseed protein isolates as valuable ingredients in the development of sustainable and high-quality food formulations [21]. Their incorporation not only reduces reliance on animal-derived proteins but also enriches formulations with bioactive compounds, including polyphenols and essential amino acids [13]. However, despite their nutritional benefits, rapeseed-derived proteins face certain technological and sensory challenges. Antinutritional factors (ANFs), such as glucosinolates and phytic acid, along with plant-specific flavour notes and limited gelation capacity, may impact product acceptability [12]. Innovative processing methods, including enzymatic modification fermentation, and the removal of anti-nutritional factors prior to protein isolation, have been investigated to enhance both functional and sensory attributes of rapeseed proteins [12,22,23]. Such advancements have considerably expanded the potential for their successful incorporation into complex food systems. Although the development of food-grade rapeseed protein isolates is challenged by the complexity and cost of processing, both factors must be considered within the broader context of the rapeseed value chain, where each processing step contributes to overall resource efficiency and value creation, particularly in light of the growing demand for rapeseed oil and the increasing production volume of rapeseed in the European Union [24]. The feasibility of rapeseed by-product valorization has been demonstrated through several successful industrial implementations. The Polish company NapiFeryn BioTech has developed a rapeseed protein concentrate obtained from press cake, which has received EFSA approval as a novel food, confirming its safety for human consumption [25]. In addition, commercially available protein isolates such as Vertis™ CanolaPRO (DSM, The Netherlands), derived from rapeseed oil processing by-products and recognized as “Upcycled Certified”, highlight not only the economic viability of rapeseed proteins but also their contribution to reducing food loss, promoting circular biomass use, and providing nutritionally complete, high-quality proteins suitable for diverse food applications.

Chicken pâté, a traditional spreadable meat product, holds nutritional and sensory significance within the category of emulsified poultry foods [26]. It is typically made from a blend of finely ground liver or meat, fat, and selected spices, often with the addition of functional ingredients to enhance flavour, consistency, and stability [27]. The texture of chicken pâtés, which may range from silky and uniform to coarser or firmer variants, depends greatly on the meat and fat composition and structure, influencing key sensory parameters such as creaminess, juiciness, and mouthfeel [28]. Beyond the primary raw materials, the incorporation of hydrocolloids, such as plant-based protein isolates are of great importance, as they substantially affect the textural, rheological, and storage-related characteristics of the product [3,13,27].

Driven by the need for nutritionally improved and technologically reliable formulations, the present study conducted a comparative quality evaluation of chicken pâtés prepared with either SPI or RPI, in combination with sunflower or rapeseed oil. Given that chicken pâtés are emulsion-based products highly sensitive to both protein and fat composition, particular emphasis was placed on textural and rheological properties, including firmness and work of shear, as these parameters provide essential insights into structural integrity, handling characteristics, and consumer-relevant quality throughout storage [27,28,29]. Building on these considerations regarding protein and fat composition, the study also focused on nutritional aspects. Shifting the fatty acid profile by replacing saturated fatty acids (SFA) with monounsaturated (MUFA) and polyunsaturated fatty acids (PUFA) through the substitution of sunflower oil with rapeseed oil was assessed as a promising strategy to meet nutritional guidelines. These guidelines emphasize reducing SFA intake while achieving a PUFA/SFA ratio above 0.4 and maintaining an n-6/n-3 ratio below 4, which are parameters widely recognized as indicators of high fat quality [30,31,32]. Consequently, enhancing the fatty acid profile of pâtés, as commonly consumed food products may support more favourable lipid profiles and help reduce the risk of metabolic disorders [33,34].

Motivated by these considerations, the present study adopts a comprehensive approach to assess the combined effects of protein and fat sources on the technological, nutritional, textural, rheological, and sensory properties of chicken pâtés. This multidimensional evaluation enabled a thorough investigation of the feasibility of rapeseed proteins as functional alternatives to traditional soybean proteins, while also elucidating their influence on product stability, textural integrity, and consumer-relevant attributes, thereby providing insights into the development of sustainable, nutritionally enhanced, and widely acceptable meat products.

## 2. Materials and Methods

### 2.1. Production of Rapeseed Protein Isolate (RPI)

RPI used in this study was obtained according to the procedure described by Đermanović et al. [22], but produced on a larger scale in a pilot plant within the Institute of Food Technology in Novi Sad. The protein isolation process was carried out in a specially designed reactor system that enabled all necessary operations to be performed under controlled and continuous conditions. The first step was dephenolization, during which the material was treated with 84% ethanol for 30 min at room temperature to remove phenolic and other antinutritional compounds. After this treatment, the liquid fraction was removed using an integrated phase separation system, while the solid residue remained in the reactor for further protein extraction. Deionized water was then added at a solid-to-liquid ratio of 1:20 (*w*/*v*), and the pH was adjusted to 12 with 1 M NaOH to solubilize the proteins. The mixture was stirred for 60 min at room temperature. Proteins were subsequently precipitated by adjusting the pH to 4.5 using 1 M HCl and recovered by centrifugation at 4000 rpm for 20 min at 4 °C (Eppendorf Centrifuge 5910 Ri, Hamburg, Germany). The resulting precipitate was washed, and the obtained final product was freeze-dried using Alpha 1–4 LDplus (Martin Christ, Osterode am Harz, Germany) at 0.0128 bar for 24 h before storage, having a protein content of at least 86.0% on a dry matter basis, determined using the Kjeldahl method [35] (ISO 937), with amino acid composition presented in Appendix A. The use of rapeseed cake as a raw material, combined with the application of a specially developed pilot reactor, provides a sustainable and scalable route for the production of high-purity rapeseed protein isolates with favourable techno-functional properties, making them suitable for further application in food products.

### 2.2. Formulation and Processing of Pâté Samples

Fresh chicken breast, drumstick, and liver were obtained from a local supplier (Perutnina Ptuj-Topiko LLC, Bačka Topola, Serbia), and their proximate composition is provided in Appendix A. Chicken breast, drumstick, and thigh meat were cut into 2.5 cm cubes and cooked in water at temperatures below 80 °C for 30 to 40 min. The resulting broth was separated and maintained at 60 °C until use. Pâté preparation was performed using a Thermomix^®^ TM5 (Vorwerk SE & Co. KG, Wuppertal, Germany). Chicken liver was raw homogenized with a half of the defined quantity of salt for the given formulation, until a uniform mass was obtained. Pre-cooked chicken meat was roughly homogenized at low speed. The sunflower oil (Dijamant LLC, Zrenjanin, Serbia) or refined rapeseed oil (Expur S.A., Slobozia, Romania) (fatty acid composition shown in Appendix A) was added and all together were heated above 65 °C. Homogenization continued at increasing speed while the remaining salt and either SPI Soypro 900E (protein content ≥90.0%; Alimenta, Novi Sad, Serbia) or the obtained RPI were added. Hot broth was slowly added during homogenization to maintain the mixture temperature around 60 °C, with the final temperature decreasing to approximately 45 °C. At the end of homogenization, the pre-homogenized liver and the rest of the ingredients were incorporated and mixed until uniform distribution was achieved. The pâté mass was filled into 100 mL glass jars with metal lids. Thermal treatment was performed in a vertical autoclave (TERRA Food-Tech^®^ CFS-75V, Barcelona, Spain) until the core temperature reached 80 °C and was maintained for 3 min. After cooking, samples were cooled in cold water and stored at 4 °C until analysis. As shown in Table 1, four pâté formulations (F1–F4) were prepared by varying the type of protein isolate (soy or rapeseed) and oil (sunflower or rapeseed). The amounts of other ingredients were identical in all formulations.

### 2.3. Sampling and Storage Conditions

For sampling, three randomly selected pâtés (in glass jars) were taken from each batch after 0, 30, 45, and 60 days of refrigerated storage at +4 °C. All analyses were performed on the day of sampling and conducted in duplicate for each sample.

### 2.4. Physicochemical Analyses

The chemical composition of the pâtés was determined using the standard ISO methods. Moisture content was determined according to ISO 1442 (1997) [36], protein content (calculated as Kjeldahl N × 6.25) following ISO 937 (1978) [35], and fat content following ISO 1443 (1973) [37]. Ash content was measured according to ISO 936 (1998) [38]. Carbohydrate content was calculated by difference [39].

The pH of the pâté samples was measured using a Testo 205 digital pH meter (Testo SE & Co. KGaA, Titisee-Neustadt, Germany), equipped with a combined penetration electrode and temperature probe. Water activity (a_w_) was determined with a LabSwift-aw meter (Novasina AG, Lachen, Switzerland). Measurements were performed at room temperature following the manufacturer’s guidelines.

### 2.5. Colour Analyses

Colour measurements were conducted on the pâté samples using a portable colorimeter (CR-400, Konica Minolta, Tokyo, Japan) equipped with a light protection tube (CR-A33b). Measurements were taken at a 0° viewing angle using an 8 mm aperture, illuminant D65 and pulsed xenon lamp as a default light source [40]. The instrument was calibrated before each session using standard white tile provided by the manufacturer. The colour was quantified in the *CIEL*a*b** colour space, recording lightness (*L**), redness (*a**), and yellowness (*b**) values. Three readings were taken for each sample at representative, homogeneous points.

### 2.6. Textural and Rheological Analyses

The textural properties of the pâtés were evaluated using a Texture Analyzer TA.XT Plus (Stable Micro Systems, Godalming, UK) following the Margarine Spreadability test (method MAR4_SR.PRJ). The test employed the Spreadability Rig HDP/SR, which consists of an upper conical probe fixed to a metal platform, maintaining a constant distance of 23 mm above a conical cup serving as the static base. The rig was calibrated to hold the upper component at 30 mm height. The test parameters included a 5 kg load cell, a test temperature of 22 °C, with a pre-test speed of 1.0 mm/s, test speed of 3.0 mm/s, and post-test speed of 10.0 mm/s [27,41]. Force applied during probe penetration was recorded until the maximum penetration depth was reached. Hardness was assessed based on the force values at specific penetration depths, while the work of shearing was calculated from the area under the force-time curve.

Rheological measurements were performed using HAAKE MARS rheometer (Karlsruhe, Germany) equipped with a serrated parallel plate geometry (d = 35 mm, gap = 1 mm). Prior to the measurements, the samples were allowed to rest for 10 min in order to release residual stress induced by lowering plate geometry. To investigate viscoelastic behaviour, i.e., sample structure and its changes over storage time, dynamic oscillatory measurements were performed. Stress sweep tests, carried out as preliminary tests to determine the viscoelastic region (LVR) were conducted by varying shear stress for 1 to 1000 Pa at a constant frequency of 1 Hz. Subsequently, frequency sweep tests were determined at constant shear stress value (previously determined from stress sweep tests), over a frequency range 1 to 10 Hz and obtained elastic (G′) and viscous (G″) moduli were recorded. All tests were conducted at 22 °C in triplicate.

### 2.7. Fatty Acid Profile and TBARS Analyses

The total lipids from pâté samples were extracted by cold extraction according to the method described by Folch et al. [42], using a chloroform–methanol mixture (2:1, *v*/*v*) for 2.5 h. The extracts were then filtered and evaporated to dryness in a vacuum dryer (Binder VD 115, Tuttlingen, Germany) to remove residual solvents. The obtained oils were flushed with a stream of nitrogen to prevent oxidation. The preparation of fatty acid methyl esters (FAMEs) followed immediately after extraction, in accordance with ISO 12966-2:2017 [43]. Gas chromatographic analysis was carried out using an Agilent 7890A system (Agilent Technologies, Santa Clara, CA, USA) equipped with a flame ionization detector (FID). FAME separation was performed on a fused silica capillary column SP-2560 (100 m × 0.25 mm, 0.20 μm film thickness; Supelco, Bellefonte, PA, USA). The injector and detector were both maintained at 250 °C, helium was employed as the carrier gas, and 1 μL of sample was injected in split mode (1:30). The oven was initially held at 140 °C for 3 min, then programmed to rise to 220 °C at 3 °C/min (with a 5 min hold), and finally increased to 240 °C at 2 °C/min (with a 10 min hold). Identification of the individual esters was achieved by comparing retention times with those of the reference standard mixture (Supelco 37 Component FAME Mix, Sigma-Aldrich, St. Louis, MI, USA), which contains 37 fatty acids. Quantification was performed by peak area comparison, and fatty acid composition was expressed as grams of fatty acids per 100 g of sample.

TBARS (2-thiobarbituric acid reactive substances) test was performed according to Šojić et al. [40]. TBARS values were expressed as milligrams of malondialdehyde per kilogram of pâté (mg MDA·kg^−1^ sample).

### 2.8. Sensory Analyses

The first step in developing a sensory lexicon for chicken pâté was collecting and selecting relevant terminology from scientific literature, encompassing a broad range of sensory characteristics specific to this product. The literature review was conducted using Scopus, Web of Science, and Google Scholar, with keywords focused on sensory evaluation of meat products, particularly appearance, odour, aroma, taste, texture, and mouthfeel of poultry-based foods. A sensory expert team from the Institute of Food Technology in Novi Sad analysed the sources, refining existing terms and introducing new ones to reflect the unique characteristics of chicken pâté. Selected descriptors were clearly defined, non-evaluative, unambiguous, and non-redundant, and then translated into the native language of the panellists to ensure consistent and precise evaluation.

Before conducting the lexicon-sorting step, sensory experts provided detailed explanations and training on each of the selected sensory descriptors to ensure a common understanding among panelists. The lexicon sorting was carried out using the Rate-All-That-Apply (RATA) method. Chicken pâté samples, labelled with three-digit codes, were evaluated at room temperature (22 ± 2 °C) after 30 min of equilibration from refrigerated storage (+4 °C). Each RATA session lasted 90 min and was divided into two 45 min sub-sessions. Panelists assessed a set of four pâté formulations (one per treatment). Sample presentation and attribute order were randomized using a balanced block design. Panellists rated each perceived attribute on a 5-point scale (1 = low, 3 = medium, 5 = high), selecting only the descriptors they noticed. A 60 s pause with mouth rinsing using mineral water was included between samples. Attributes selected by at least 30% of panellists were included in the descriptive analysis (DA).

Prior to the descriptive analysis (DA), panelists underwent structured training over three 90 min sessions in one week. The training included familiarization with commercially available chicken pâtés, alignment on descriptor concepts with reference standards and intensity examples for sensory descriptors, and exercises to build consensus and practice evaluating attributes on a 7-point scale. Consistency of understanding was verified through group discussions and calibration exercises, resolving any deviations before the formal DA evaluation.

For the DA, four chicken pâté samples were evaluated by a panel of 10 trained assessors from the Institute of Food Technology, Novi Sad, in two 90 min sessions, following a balanced block design, with each descriptor rated on a 7-point scale. The descriptors retained for the descriptive analysis included: Appearance—colour homogeneity; colour was determined visually using the NCS System^®^, and the obtained colour codes were S1515-Y10R, S2010-Y10R, S2020-Y20R, S1515-Y10R, S1515-Y10R, S2010-Y10R, and S2020-Y20R; Tactile texture—cohesiveness, spreadability; Orthonasal odour—odor intensity, meaty odour, broth odour, musty odour, soy isolate odour, rapeseed isolate odour, coriander odour, liver odour; Taste—salty, umami, bitter, sour; Flavour/aroma/trigeminal sensation—overall flavour intensity, flavour fullness/balance, flavour persistence, pungency, irritation/tingling, refreshing effect, chicken flavour note, broth flavour, liver flavour, coriander flavour, soy isolate flavour, rapeseed isolate flavour; Aftertaste—aftertaste/odour, and Texture/Mouthfeel—smoothness, creaminess, graininess, stickiness, moistness, and fatty/oil feel.

This study was approved by the Ethics Committee of the Institute of Food Technology in Novi Sad, University of Novi Sad, Serbia (Ref. No. 175/I/23-3).

### 2.9. Statistical Analysis

Factorial ANOVA was performed in Statistica 14.0.0.15 (TIBCO Software Inc., Palo Alto, CA, USA) in order to analyse the effects of formulation type and storage time on the measured physicochemical parameters, with Duncan’s post hoc test applied for mean comparisons. Differences were considered statistically significant at *p* < 0.05.

Statistical processing of the sensory evaluation data was performed using XLSTAT software (version 2023.3.1), including analysis of attribute frequencies derived from the RATA method, one-way ANOVA on intensity scores, Tukey post hoc comparisons, and principal component analysis (PCA) to explore patterns and relationships among the main sensory variables of all samples.

## 3. Results and Discussion

### 3.1. Proximate Composition

The proximate composition of chicken pâté formulations is presented in Table 2, showing that differences among samples were generally modest due to the common meat base, while some variations were observed depending on the protein isolate and oil used. Considering that F1 represents the conventional formulation and served as the control, it was particularly interesting to compare how the alternative replacements of oil and protein in F2–F4 influenced the overall composition of the products. Moisture ranged from 51.9% (F3) to 53.9% (F2), with F3 showing significantly lower (*p* < 0.05) values compared to the other treatments, which may be attributed to the lower water-binding capacity of RPI compared to SPI, as well as minor differences in fat content and matrix density that may have influenced water retention within the pâté structure [27,29]. This result aligns with previous findings, indicating that rapeseed protein exhibits lower water-holding but improved oil-holding capacity compared to soy protein, thereby influencing moisture distribution and textural attributes in protein-based food systems [9]. Fat content varied slightly among the formulations, with statistically significant differences observed between pâtés containing different protein isolates. Formulations with RPI (F3 and F4) consistently exhibited higher fat content, regardless of the type of oil used. This suggests that the oil phase may have been more effectively retained within the matrix during thermal processing, likely due to stronger protein–lipid interactions associated with RPI [9,10]. Higher fat content observed in F3 is consistent with its lower moisture level, highlighting the inverse relationship between water and lipid fractions. This observation aligns with previous findings in commercial and reformulated pâtés, where lipid level is a major determinant of caloric value and textural properties [28,44], and is consistent with reports that rapeseed protein exhibits superior oil-holding capacity compared to soy protein [9]. Protein values ranged from 11.8% to 12.2%, with SPI-based samples (F1, F2) maintaining slightly higher protein contents than their RPI counterparts, consistent with the higher nominal protein fraction of soy isolates. Comparable results were reported by Trindade et al. [29], who showed that reformulation with pea protein isolate maintained protein levels close to conventional pâté. Ash content varied from 1.61% (F2) to 1.93% (F1). Although differences among treatments were minor, higher ash values may be linked to differences in the mineral content of the raw materials used [45,46]. Similar trends were observed in chicken liver pâté formulations with hydrocolloid additions, where ash content remained relatively stable despite formulation changes [27]. Carbohydrate levels, calculated by difference, were relatively low across all pâté formulations. Although F3 and F4 exhibited slightly higher values, these variations primarily reflect the inverse relationship with moisture and lipid content rather than any direct effect of the protein isolate. This pattern is consistent with typical proximate composition analyses, where minor fluctuations in calculated carbohydrate content largely result from changes in other major constituents (moisture, protein, fat, and ash) [39].

### 3.2. pH and Water Activity (a_w_)

The pH of meat emulsions, including pâté-type products, is a critical parameter that affects protein functionality, and consequently water-holding capacity, texture, emulsion stability, and microbial safety [47,48]. Maintaining pH within an optimal range is essential for ensuring desirable texture, sensory attributes, and microbiological safety during storage. In the present study, pH values of the chicken pâté formulations (Table 3) ranged from 5.96 to 6.42, which falls within the typical range of approximately 5.8 to 6.9 reported for spreadable poultry-based pâtés [27,49,50]. During storage, all formulations showed a decrease in pH in comparison to day 0, with statistically significant effects of storage time, formulation, and their interaction (*p* < 0.05). At the beginning of storage, formulations F1 and F2 exhibited higher pH values compared to F3 and F4, reflecting differences in buffering capacity between protein sources. This observation is consistent with previous reports indicating that protein composition and functional properties influence pH behaviour in meat emulsions [51,52].

Composition of proteins, lipids, and carbohydrates directly affects water retention, as these macromolecules bind free water to varying degrees, thereby influencing product texture and overall stability [53,54]. In the present study, a_w_ values (Table 3) remained stable during 60 days of storage for all formulations, ranging from 0.93 to 0.94, with no statistically significant effects of storage time, formulation, or their interaction (*p* > 0.05). This demonstrates that both soy and rapeseed proteins effectively retained water within the pâté matrix throughout the storage period, ensuring physicochemical stability. Remarkably, RPI, despite its lower protein content, exhibited comparable technological functionality to SPI when incorporated at the same proportion, effectively contributing to water retention and emulsion stability. These results indicate that RPI can serve as a functional alternative to SPI in pâté-type products without compromising emulsion stability or water-holding capacity.

### 3.3. Colour Characteristics

Changes in colour parameters differed significantly (*p* < 0.05) among the pâté samples (F1–F4), being influenced by storage time and formulation type, as well as their interaction (Figure 1). At day 0, SPI-based pâtés (F1 and F2) exhibited higher lightness (*L**) values (*p* < 0.05), then RPI-based samples (F3 and F4). This initial difference is likely associated with the intrinsic pigments present in rapeseed protein isolate, which contribute to a lower *L** and higher *b** (yellowness) [9]. Over the 60-day storage period, the lightness of F3 and F4 gradually increased as a result of brightening, most likely due to pigment, lipid and protein oxidation, as well as enzymatic degradation and structural changes [55,56,57]. On the other hand, this colour characteristic remained almost the same in F1 and F2. Thus, the initial differences between SPI and RPI formulations became less pronounced after two months of refrigerated storage. Regarding yellowness (*b**), RPI-containing pâtés showed significantly higher initial values, consistent with the presence of natural pigments in rapeseed protein [9,10]. During storage, *b** values in all formulations tended toward similar levels, indicating a convergence in colour appearance over time, which can be attributed to pigment degradation and the stabilizing influence of the meat matrix, as observed for the *L** parameter. Although the RPI formulations initially exhibited higher redness, i.e., higher *a** values, this parameter gradually decreased and converged with the other formulations during storage, likely reflecting pigment oxidation [10] and interactions within the protein–oil–meat matrix.

Colour changes during storage are frequently observed in meat emulsions and may reflect either stabilization or degradation of colour, depending on formulation and storage conditions. Therefore, it is essential to monitor the overall impact of all colour changes (*L**, *a**, and *b**) through the total colour difference (Δ*E**), which provides a comprehensive measure of perceptible changes and highlights their significance for the product’s visual appearance [58,59]. Among the formulations, F3 showed the highest Δ*E** (10.5), reflecting the most pronounced overall colour change, whereas F4, containing both RPI and rapeseed oil, exhibited the lowest Δ*E** (5.69), suggesting enhanced colour stability throughout storage. When compared with the conventional control (F1), F4 exhibited comparable colour parameters by the end of the storage period, indicating that the combination of rapeseed-derived ingredients preserves a visual appearance similar to the standard formulation.

### 3.4. Textural and Rheological Behaviour

At the beginning of storage, formulations containing RPI (F3 and F4) exhibited higher firmness and shear resistance compared to their soy protein counterparts (F1 and F2) (Table 4). This demonstrates that RPI facilitated the formation of a denser and more cohesive protein–fat network, resulting in improved initial structural integrity. These observations are consistent with the proximate composition results, particularly for F3, where lower moisture and higher fat content likely contributed to a more compact matrix with less free water available for mobility. The higher fat fraction, more efficiently retained due to the superior oil-holding capacity of RPI [9,10], reinforced the protein–lipid network, resulting in firmer texture and reduced spreadability [28].

During 60 days of refrigerated storage, both firmness and work of shear generally increased across all formulations, reflecting progressive structural reinforcement, most likely due to water redistribution and gradual protein network rearrangements. This effect was most pronounced in formulation F3, where the increases in both parameters were consistently significant over time, confirming the stabilizing role of RPI and its capacity to sustain textural integrity during extended storage. From a practical perspective, this finding suggests that comparable or even improved product stability can be achieved with lower amounts of protein isolate when RPI is used, offering a potential economic advantage in formulation. Amaral et al. [60] also reported that the texture parameters of lamb pâté, such as hardness, adhesiveness, and cohesiveness, were also affected by storage time.

The rheological behaviour of the pâtés, assessed through storage modulus (G′) and loss modulus (G″), provided complementary insights into their structural dynamics during storage (Figure 2). Across formulations, G′ generally reflected the elastic, solid-like character of the matrix, while G″ indicated the viscous, energy-dissipating component. At day 0, formulations containing RPI (F3 and F4) exhibited higher G′ values compared to SPI-based formulations, consistent with the initial firmness and work of shear results, confirming the contribution of rapeseed protein to a denser, more elastic network.

However, during storage, a gradual decline in G′ was observed, particularly by day 60, which contrasts with the increasing trend in firmness and work of shear. This difference can be explained by the distinct physical principles underlying the measurements: texture analysis quantifies resistance to localized deformation under compression or cutting, whereas rheological measurements assess the global viscoelastic response of the matrix under oscillatory stress [61]. Moreover, the obtained difference between textural and rheological behaviour might also be influenced by microstructural changes occurring during storage period. Namely, progressive aggregation of protein and rearrangement of the fat-protein matrix result in a network that is more compact but less elastic. While this microstructural tightening actually increases the structure’s resistance to localized forces, leading to greater firmness, it simultaneously hinders its capacity for recovery under low-amplitude oscillatory stress. Consequently, this leads to a lower storage modulus (G′) values. Therefore, while water redistribution and protein network rearrangements may increase local firmness and shear resistance, they can simultaneously lead to slight softening of the overall elastic network detectable in oscillatory rheology. According to Sadeghi-Mehr et al. [62] large-deformation tests such as textural measurement does not necessarily correlate with those from small-deformation tests such as dynamic oscillatory measurements. Namely, Broucke et al. [63] revealed that addition of texturized soy proteins into lean chicken meat batter system resulted in lower final G’ values due to discontinuities in the meat matrix, but higher hardness values. However, both sets of data converge on the conclusion that RPI enhances the structural stability of the pâté matrix, maintaining textural integrity and providing a more elastic network compared to SPI formulations. The combination of textural and rheological measurements thus offers a comprehensive understanding of pâté quality, highlighting the functional potential of rapeseed protein for improving both consumer-perceived texture and product stability during storage.

### 3.5. Fatty Acid Composition and Oxidative Stability

Given the critical role of oil fatty acid composition, particularly regarding cardiovascular and metabolic outcomes in human health, this study also aimed to investigate the potential of replacing sunflower oil with rapeseed oil to enhance the nutritional quality of chicken pâtés. Saturated fatty acids are associated with elevated LDL-cholesterol levels, whereas their replacement with cis-monounsaturated (MUFA) and cis-polyunsaturated (PUFA) fatty acids improves lipid metabolism and reduces the risk of metabolic disorders [33,34]. Rapeseed oil contains less than 8% saturated fatty acids and provides a more favorable n-6/n-3 ratio (≈2:1) compared to sunflower oil, which is dominated by linoleic acid (C18:2n-6) and lacks α-linolenic acid (C18:3n-3). The higher α-linolenic and oleic acid contents in rapeseed oil contribute to reduced LDL oxidation, improved lipid profile, and anti-inflammatory effects, supporting its role as a nutritionally superior fat source [30,32,64].

In parallel, the substitution of soy protein with rapeseed protein was evaluated to assess the combined effects of these modifications on the overall fatty acid profile and nutritional quality of the products. Hence, the replacement of sunflower oil with rapeseed oil (formulations F2 and F4) markedly improved the fatty acid profile of the pâtés (Table 5). Oleic acid (C18:1n-9) increased from 9.03% in the sunflower oil control (F1) to 18.1% in F2 and 18.7% in F4, reflecting the higher proportion of MUFAs characteristic of rapeseed oil. Linoleic acid (C18:2n-6), the dominant PUFA in sunflower oil, decreased from 16.6% in F1 to 5.91% in F2 and 6.29% in F4, resulting in a more balanced fatty acid composition with increased MUFA content and a reduced relative proportion of n-6 PUFAs. The most nutritionally significant change was observed for α-linolenic acid (C18:3n-3), which increased from negligible levels in F1 (0.09%) and F3 (0.11%) to 2.27% in F2 and 2.32% in F4, the rapeseed oil formulations. This resulted in a marked improvement in the n-6/n-3, decreasing from an unfavourable 159 in F1 to 2.7 in F2 and 2.8 in F4, thereby aligning with recommended values and promoting cardiovascular and metabolic health.

From a broader perspective, these findings confirm that rapeseed oil is a highly effective reformulation strategy for improving the nutritional quality of meat products. Comparable conclusions were drawn by Bilska and Krzywdzińska-Bartkowiak [65], who reported that rapeseed oil incorporation in liver pâtés provided the most favourable n-6/n-3 ratio among tested vegetable oils, while also maintaining oxidative stability. Considering the limited literature specifically addressing rapeseed oil application in meat emulsions, the present results provide valuable evidence supporting its use as a functional ingredient to improve both the nutritional and technological quality of poultry pâtés. In particular, the formulations containing rapeseed oil (F2 and F4) can be classified as products rich in n-3 and monounsaturated fatty acids, in accordance with Regulation (EC) No 1924/2006 and Commission Regulation (EU) No 116/2010, which establish the criteria for nutrition claims [66]. Such labelling potential was not observed in the sunflower oil-based samples, highlighting the superior health-promoting properties of rapeseed oil. Beyond the improvement of the fatty acid profile, the incorporation of rapeseed oil enables the development of functional meat products with added nutritional value, thereby enhancing their market competitiveness while simultaneously contributing to the promotion of healthier alternatives within the processed meat sector.

Lipid oxidation is a critical factor affecting the quality, shelf-life, and sensory attributes of meat products [67,68]. Therefore, monitoring TBARS (2-thiobarbituric acid reactive substances) in this study provided insight into how different protein isolates and lipid sources influence the oxidative stability of chicken pâtés. Storage time and formulation type, as well as their interaction significantly affected lipid oxidation (*p* < 0.05) in the tested pâtés (Table 6).

At the beginning of storage (day 0), TBARS values were comparable among formulations (1.37–1.41 mg MDA/kg). Over time, changes varied depending on the formulation. The control (F1) showed a slight increase until day 30, followed by a significant decline to 1.13 mg/kg at day 60 (*p* < 0.05). F2 remained relatively stable (*p* > 0.05), whereas RPI-based formulations (F3 and F4) exhibited significantly lower TBARS at later stages. F3 decreased from 1.51 mg/kg (day 30) to 0.66 mg/kg at day 60, while F4 decreased from 1.46 mg/kg (day 30) to 0.75 mg/kg at day 60, corresponding to reductions of ~56% and ~49%, respectively. The significant reduction in TBARS after 45–60 (*p* < 0.05) days may reflect MDA’s binding to protein amino groups, forming stable MDA–lysine adducts, a phenomenon well documented in lipoxidation chemistry [69,70] and potentially enhanced by the high lysine content of rapeseed proteins [22], whose reactive amino groups readily bind MDA, thereby contributing to the pronounced decrease in measurable TBARS values.

This “rise-and-fall” pattern has been reported in different pâtés enriched with natural antioxidants [59,71,72], while Carpes & Haminiuk [73] also observed reduced TBARS values in products containing grape seed and walnut extracts. Evidence on antioxidant peptides [74,75] and the protective role of oleic acid-rich oils [74,75] indicates that bioactive compounds (peptides, oleic acid-rich lipids, and plant-derived polyphenols) may contribute to lipid stabilization during storage. Rapeseed oil, rich in oleic acid (MUFA), reduced oxidation compared to sunflower oil, which is high in linoleic acid (PUFA) [65,76]. Consistent with Bilska et al. [65], this study observed that formulations incorporating rapeseed oil demonstrated superior oxidative stability. This aligns with the reported benefits of replacing animal fat with vegetable oils, particularly rapeseed oil, in enhancing the oxidative stability and sensory quality of liver pâté-type processed meats. Rapeseed protein further enhanced stability, likely through antioxidant peptides that scavenge free radicals and bind reactive carbonyls [74,75]. All formulations remained well below 2 mg MDA/kg, the threshold for perceptible rancidity [77] confirming that chicken pâtés retained acceptable sensory quality throughout 60 days. Overall, the combined choice of protein and oil significantly influenced oxidative stability, underscoring the importance of tailored formulation strategies.

### 3.6. Sensory Evaluation

The initial sensory evaluation of chicken pâtés F1–F4 revealed significant differences in 25 of the 38 assessed attributes (*p* < 0.05), indicating that panellists could reliably discriminate between the formulations. The most pronounced distinctions were observed in odour and aroma notes linked to the type of protein isolate, particularly rapeseed protein (*p* < 0.001), as well as in aftertaste intensity (*p* < 0.0001), overall aroma/flavour intensity (*p* = 0.001), flavor persistence (*p* = 0.010), pungency (*p* = 0.003), liver aroma (*p* = 0.034), soy protein aroma (*p* < 0.0001), and perceived fattiness (*p* = 0.024). These results demonstrate that the choice of protein isolate and oil significantly shaped the dominant sensory characteristics. Visual attributes were also influenced, with significant differences in colour shades (*p* < 0.0001, NCS System^®^), confirming that formulation impacted both olfactory/gustatory properties and appearance. PCA and factor analysis at day 0 highlighted a clear separation of sensory profiles. Sample F4 was strongly associated with rapeseed-related aroma, persistent aftertaste, spiciness, and overall flavour intensity, whereas F3 was characterized by creamy texture, mild rapeseed aroma, and a specific colour nuance (S1010-Y20R). F1 and F2, both containing SPI, were linked to soy-derived aromas, with F2 exhibiting a more refreshing profile and less dominant overall flavour. Textural attributes contributed moderately to differentiation, with granularity, stickiness, and smoothness influencing the profiles to a lesser extent than odour and flavour.

Afterwards, sensory analysis of chicken pâtés (F1–F4) was performed on days 45 and 60, whereas physicochemical analyses also included day 30. While chemical composition often changes early during storage, sensory attributes, i.e., aroma, taste, and texture tend to exhibit more pronounced variations at later stages, especially when volatile compounds accumulate or degradation products reach sensory thresholds. Several studies on meat and meat-derived products support this sequential trend, where physicochemical indices shift first, followed by perceivable changes in sensory quality [78,79]. The relatively large sample quantities required for sensory testing further guided the timing of these analyses. By aligning sensory evaluations with physicochemical outcomes, assessments at days 45 and 60 captured changes most relevant to consumer perception and the product’s intended shelf life.

Analysis of storage effects between days 45 and 60 indicated that most sensory attributes remained stable (*p* > 0.05), with the exceptions of aftertaste intensity (*p* = 0.004) and irritation (*p* = 0.038). Aftertaste intensity declined further, disappearing entirely in F1 and F3, while F4 retained a minimal but perceptible aftertaste, reflecting better preservation of its aromatic profile. The slight reappearance of irritation in F2 and F3 suggests the formation of new trigeminal-active compounds, likely arising from minor lipid or protein degradation. Other attributes, including texture and oil-related aroma, remained largely unchanged, although small trends were observed: F2 showed a slight increase in soybean-related aroma, whereas F1 and F3 exhibited minor reductions in creaminess and aroma fullness. Notably, F4 maintained a stable profile in both texture and aroma, indicating a formulation resilient to storage-induced changes. The obtained findings could be related to instrumental textural measurements, where RPI-based samples (F3 and F4) showed higher firmness and work of shear values and thus more cohesive protein-fat matrix than SPI-based samples (F1 and F2). During the storage period, the increase in textural parameters was observed suggesting progressive structural reinforcement which could be related to protein network rearrangement and water redistribution. This finding is in agreement with Amaral et al. [60] who also revealed that hardness of pate increase during storage period due to emulsion destabilization, with separation of water and fat of protein matrix. Both, textural and sensory analysis confirmed that rapeseed protein isolate addition resulted in a stable and elastic matrix development that sustains desirable mouthfeel and textural integrity, whereas the sample containing rapeseed proteins as well as rapeseed oil showed the highly acceptable sensory profile.

For the purposes of this study, the sensory data were summarized in a single cumulative PCA biplot, integrating results from all storage times (Figure 3). This approach allows a comprehensive visualization of temporal trends across formulations. The first two principal components jointly explained 43.42% of the total variability. PC1 was primarily driven by aftertaste, stickiness, graininess, and broth flavour intensity, whereas PC2 was associated with coriander, irritation, bitterness, and aroma balance. The negative displacements along PC1 and PC2 reflected a general decline in desirable aroma attributes and the appearance of slight off-notes, including acidity, bitterness, and soy protein-related odour. Sample F3 remained relatively stable throughout storage, whereas F4 exhibited minimal displacement, confirming its robustness in preserving sensory characteristics.

Sensory evaluation of the colour of chicken pâtés F1–F4, conducted according to the NCS standard, closely corresponded with instrumental measurements. Observed colour changes were largely reflected in variations in *L**, *a**, and *b** values, with Δ*E** values aligning with perceptible differences, thereby supporting the validity of the combined approach (Figure 1). For effective monitoring of colour stability, an integrated methodology is recommended: sensory assessment captures visually relevant changes critical for consumer perception, while instrumental measurements provide objective, quantitative data for tracking trends and evaluating the effects of formulation or storage [80]. The combined application of NCS-based sensory evaluation and instrumental parameters enables precise and reproducible monitoring of colour, in accordance with both industrial and scientific standards, providing reliable information for product optimization.

Overall, these results indicate that protein type is the principal factor driving sensory differentiation in chicken pâtés, with RPI-based formulations exhibiting superior stability over 60 days. SPI-based formulations were more susceptible to aromatic and flavour deterioration, highlighting the potential need for formulation optimization, such as the incorporation of antioxidants or control of components contributing to off-flavours. Among the formulations, F4, combining rapeseed-derived ingredients, demonstrated the most balanced and persistent sensory profile, representing the most promising candidate for extended shelf-life applications.

## 4. Conclusions

This study provides one of the first demonstrations of the successful application of rapeseed protein isolates (RPI) in emulsified poultry systems, highlighting their technological and nutritional potential as sustainable alternative to conventional soybean protein isolates (SPI). The findings confirm that RPI can effectively function as a structuring and stabilizing ingredient in chicken pâtés, contributing to water binding and emulsion stability while maintaining consistent water activity (a_w_) over 60 days of storage. Although the proximate composition of the pâtés remained largely uniform due to their common meat base, statistically significant differences were observed in moisture content, lipids and proteins, reflecting the functional impact of protein source and oil type on formulation performance. Textural and rheological properties, as critical processing and handling parameters, showed that RPI, despite having lower intrinsic protein content than SPI, was able to deliver superior firmness and structural integrity in the pâtés. This indicates that good textural properties can be achieved even with reduced rapeseed protein concentration, providing opportunities for sustainable reformulation strategies. Moreover, formulations containing both rapeseed-derived ingredients exhibited enhanced oxidative stability during storage, likely due to the synergistic effect of MDA binding by rapeseed protein amino groups and the protective role of oleic-acid-rich rapeseed oil. This combination highlights clear advantages for maintaining the quality and extending the shelf life of pâté-type products. Overall, protein type was the main factor influencing the sensory characteristics of chicken pâtés, with rapeseed protein-based formulations showing superior stability over 60 days. Among them, formulation F4, combining rapeseed protein and rapeseed oil, exhibited the most balanced and persistent sensory profile, making it the most promising for extended shelf-life applications.

The results of this study demonstrate that the combined use of RPI and rapeseed oil can produce chicken pâtés that are nutritionally enhanced, technologically reliable, and sensory-acceptable. Furthermore, RPI-based formulations showed improved textural properties and greater oxidative stability, likely due to synergistic effects between rapeseed protein and oil. Replacing sunflower oil with rapeseed oil significantly improved the fatty acid profile, notably increasing α-linolenic acid (C18:3n-3) and yielding a favourable n-6/n-3 ratio (approx. 2.8). This formulation offers complementary technological and nutritional advantages, aligning with the broader trend of replacing sunflower oil with rapeseed oil in processed foods. Moreover, this study highlights the potential for sustainable reformulation by replacing SPI with RPI, which not only provides an alternative to imported soy, reducing dependency on global supply chains and associated environmental impacts, but also supports broader sustainability and regulatory objectives. Overall, this approach enables the creation of nutritionally improved and technologically robust products that contribute to circular economy principles and address consumer demand for environmentally responsible, high-quality food products.

## Figures and Tables

**Figure 1 foods-14-03841-f001:**
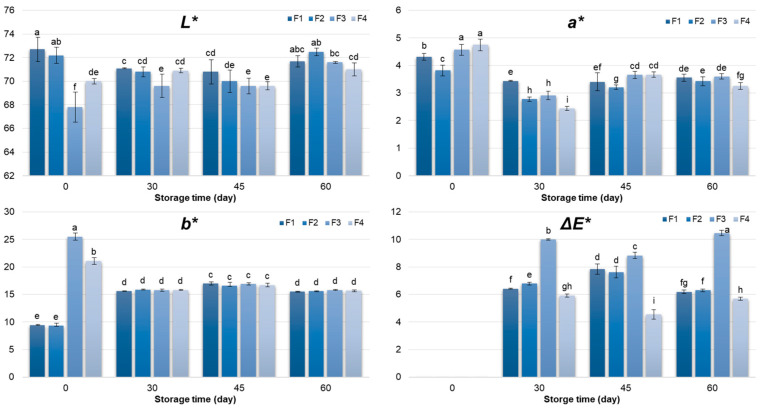
Colour characteristics (*L**, *a**, *b**, Δ*E**) of chicken pâté samples during 60 days storage period. *L**—lightness; *a**—redness; *b**—yelowness; Δ*E**—total colour difference; F1 (soy protein isolate + sunflower oil); F2 (soy protein isolate + rapeseed oil); F3 (rapeseed protein isolate + sunflower oil); F4 (rapeseed protein isolate + rapeseed oil). Different letters above the bars indicate significant differences (*p* < 0.05).

**Figure 2 foods-14-03841-f002:**
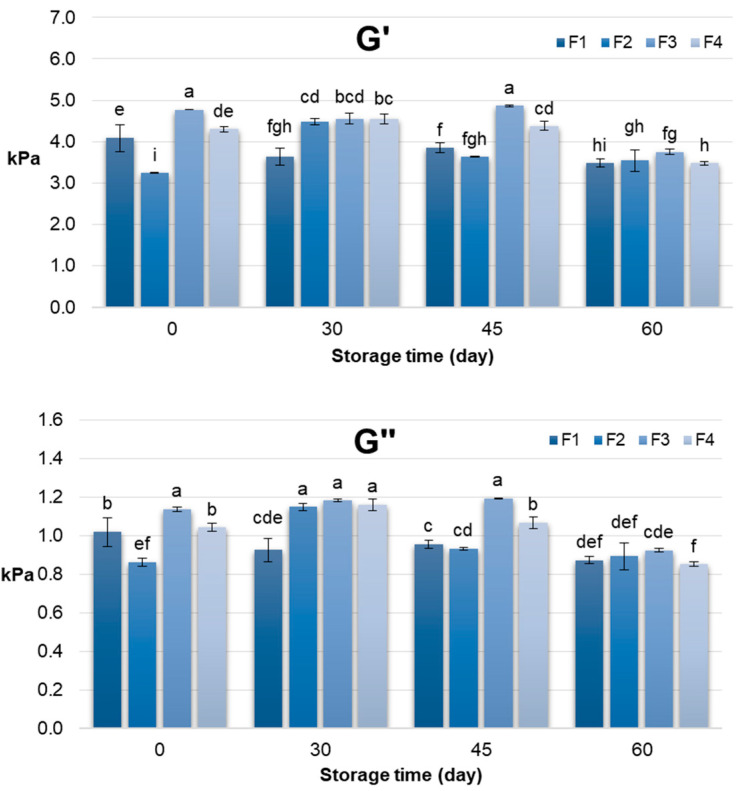
Rheological characteristics (G′ and G″) of chicken pâté samples during 60 days storage period. G′—storage modulus; G″—loss modulus; F1 (soy protein isolate + sunflower oil); F2 (soy protein isolate + rapeseed oil); F3 (rapeseed protein isolate + sunflower oil); F4 (rapeseed protein isolate + rapeseed oil). Different letters above the bars indicate significant differences (*p* < 0.05).

**Figure 3 foods-14-03841-f003:**
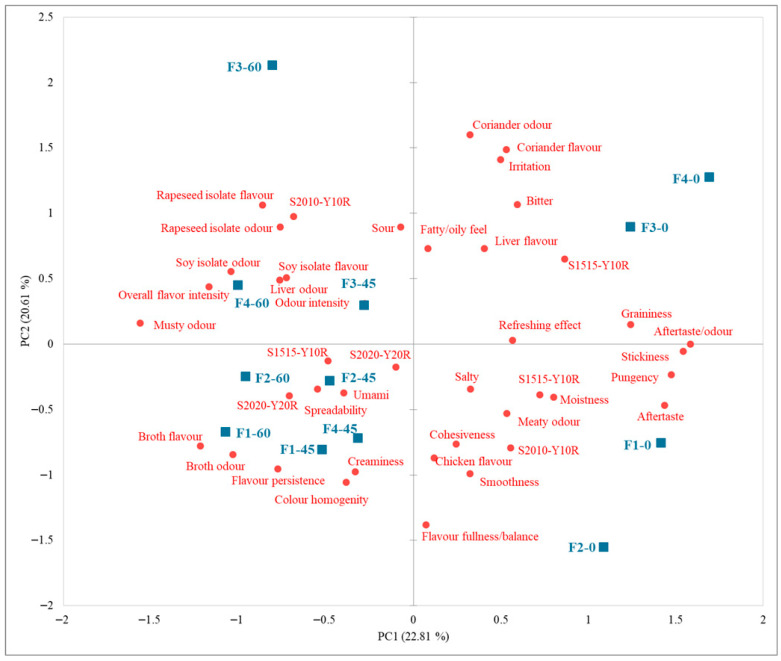
Principal Component Analysis (PCA) biplot of sensory characteristics of chicken pâté samples during 60 days storage period. F1 (soy protein isolate + sunflower oil); F2 (soy protein isolate + rapeseed oil); F3 (rapeseed protein isolate + sunflower oil); F4 (rapeseed protein isolate + rapeseed oil); PC1 and PC2—principal components explaining the variability of sensory attributes.

**Table 1 foods-14-03841-t001:** Formulation of elaborated chicken pâtés (% *w*/*w*).

Ingredient	F1	F2	F3	F4
Chicken breast and drumstick meat	30.0	30.0	30.0	30.0
Broth	26.0	26.0	26.0	26.0
Sunflower oil	28.0	-	28.0	-
Rapeseed oil	-	28.0	-	28.0
Chicken liver	10.0	10.0	10.0	10.0
Soy protein isolate	3.00	3.00	-	-
Rapeseed protein isolate	-	-	3.00	3.00
Potato starch	1.00	1.00	1.00	1.00
Onion	0.50	0.50	0.50	0.50
Coriander	0.30	0.30	0.30	0.30
Salt	1.50	1.50	1.50	1.50
Ascorbic acid	0.03	0.03	0.03	0.03

**Table 2 foods-14-03841-t002:** Proximate composition (%) of chicken pâté samples (day 0).

Analysis	Treatments			
	F1	F2	F3	F4
Moisture	53.1 ± 0.51 ^a^	53.9 ± 0.50 ^a^	51.9 ± 0.50 ^b^	53.2 ± 0.50 ^a^
Ash	1.93 ± 0.20 ^a^	1.61 ± 0.15 ^b^	1.74 ± 0.15 ^ab^	1.64 ± 0.10 ^ab^
Fat	29.5 ± 0.40 ^bc^	28.9 ± 0.40 ^c^	30.7 ± 0.40 ^a^	30.1 ± 0.40 ^ab^
Protein	12.2 ± 0.20 ^a^	12.0 ± 0.13 ^ab^	11.8 ± 0.12 ^c^	11.8 ± 0.02 ^c^
Carbohydrates	3.07 ± 0.06 ^b^	3.72 ± 0.74 ^ab^	4.10 ± 0.64 ^a^	4.04 ± 0.10 ^ab^

F1 (soy protein isolate + sunflower oil); F2 (soy protein isolate + rapeseed oil); F3 (rapeseed protein isolate + sunflower oil); F4 (rapeseed protein isolate + rapeseed oil). ^a–c^ Mean values within the same row not followed by common letter differ significantly (*p* < 0.05).

**Table 3 foods-14-03841-t003:** pH and a_w_ of chicken pâté samples during 60 days storage period.

Parameter	Treatments	Storage Time (Day)						
		0	30	45	60	T	F	T × F
pH	F1	6.40 ± 0.01 ^a^	6.22 ± 0.02 ^d^	6.20 ± 0.01 ^d^	6.28 ± 0.01 ^c^	*	*	*
	F2	6.42 ± 0.01 ^a^	6.28 ± 0.01 ^c^	6.26 ± 0.01 ^c^	6.35 ± 0.01 ^b^	*	*	*
	F3	6.08 ± 0.01 ^ef^	5.97 ± 0.01 ^i^	5.98 ± 0.04 ^hi^	6.02 ± 0.03 ^g^	*	*	*
	F4	6.09 ± 0.01 ^e^	5.96 ± 0.01 ^i^	6.01 ± 0.01 ^gh^	6.06 ± 0.02 ^f^	*	*	*
a_w_	F1	0.93 ± 0.01 ^ab^	0.93 ± 0.01 ^ab^	0.93 ± 0.01 ^ab^	0.94 ± 0.00 ^a^	ns	ns	ns
	F2	0.93 ± 0.01 ^ab^	0.94 ± 0.01 ^ab^	0.93 ± 0.01 ^ab^	0.94 ± 0.00 ^a^	ns	ns	ns
	F3	0.94 ± 0.01 ^ab^	0.93 ± 0.01 ^ab^	0.93 ± 0.01 ^ab^	0.93 ± 0.00 ^b^	ns	ns	ns
	F4	0.94 ± 0.01 ^a^	0.93 ± 0.01 ^ab^	0.93 ± 0.01 ^ab^	0.93 ± 0.00 ^ab^	ns	ns	ns

F1 (soy protein isolate + sunflower oil); F2 (soy protein isolate + rapeseed oil); F3 (rapeseed protein isolate + sunflower oil); F4 (rapeseed protein isolate + rapeseed oil); a_w_—water activity; T—time; F—formulation; T × F—interaction effect; ns = not significant; * *p* < 0.05. ^a–i^ Mean values within the same row not followed by common letter differ significantly (*p* < 0.05).

**Table 4 foods-14-03841-t004:** Textural characteristics of chicken pâté samples during 60 days storage period.

Parameter	Treatments	Storage Time (Day)						
		0	30	45	60	T	F	T × F
Firmness (N)	F1	1.32 ± 0.03 ^d^	1.42 ± 0.11 ^cd^	1.43 ± 0.10 ^cd^	1.46 ± 0.13 ^cd^	*	*	ns
	F2	1.36 ± 0.02 ^cd^	1.50 ± 0.05 ^c^	1.51 ± 0.07 ^c^	1.50 ± 0.13 ^c^	*	*	ns
	F3	1.45 ± 0.03 ^cd^	1.71 ± 0.06 ^b^	1.70 ± 0.17 ^b^	1.87 ± 0.04 ^a^	*	*	ns
	F4	1.75 ± 0.02 ^ab^	1.90 ± 0.07 ^a^	1.88 ± 0.15 ^a^	1.89 ± 0.03 ^a^	*	*	ns
Work of shear (N mm)	F1	3.62 ± 0.06 ^e^	3.88 ± 0.37 ^de^	3.91 ± 0.30 ^de^	4.24 ± 0.29 ^cd^	*	*	*
	F2	4.39 ± 0.03 ^c^	4.25 ± 0.20 ^cd^	4.18 ± 0.26 ^cd^	4.19 ± 0.32 ^cd^	*	*	*
	F3	4.48 ± 0.03 ^c^	5.08 ± 0.34 ^b^	5.15 ± 0.38 ^ab^	5.40 ± 0.28 ^ab^	*	*	*
	F4	5.19 ± 0.02 ^ab^	5.54 ± 0.07 ^a^	5.45 ± 0.15 ^ab^	5.46 ± 0.10 ^ab^	*	*	*

F1 (soy protein isolate + sunflower oil); F2 (soy protein isolate + rapeseed oil); F3 (rapeseed protein isolate + sunflower oil); F4 (rapeseed protein isolate + rapeseed oil); T—time; F—formulation; T × F—interaction effect; ns = not significant; * *p* < 0.05. ^a–e^ Mean values within the same row not followed by common letter differ significantly (*p* < 0.05).

**Table 5 foods-14-03841-t005:** Fatty acid profile of chicken pâté samples (day 0).

Fatty-Acid Composition (g per 100 g Pâté, Fresh Weight)	F1	F2	F3	F4
C14:0	0.03 ± 0.00 ^a^	0.02 ± 0.00 ^b^	0.03 ± 0.00 ^a^	0.02 ± 0.00 ^b^
C16:0	2.12 ± 0.04 ^a^	1.59 ± 0.04 ^b^	2.26 ± 0.09 ^a^	1.74 ± 0.05 ^b^
C16:1	0.07 ± 0.00 ^b^	0.10 ± 0.00 ^a^	0.08 ± 0.00 ^b^	0.12 ± 0.01 ^a^
C18:0	1.15 ± 0.08 ^a^	0.56 ± 0.03 ^b^	1.18 ± 0.07 ^a^	0.60 ± 0.10 ^b^
C18:1n-9	9.04 ± 0.12 ^c^	18.1 ± 0.16 ^b^	9.42 ± 0.11 ^c^	18.7 ± 0.18 ^a^
C18:2n-6	16.6 ± 0.13 ^b^	5.92 ± 0.06 ^c^	17.1 ± 0.13 ^a^	6.30 ± 0.13 ^d^
C20:0	0.09 ± 0.00 ^b^	0.13 ± 0.01 ^a^	0.09 ± 0.01 ^b^	0.14 ± 0.01 ^a^
C18:3n-6	Nd	0.03 ± 0.00 ^a^	Nd	0.03 ± 0.00 ^a^
C18:3n-3	0.09 ± 0.01 ^b^	2.27 ± 0.06 ^a^	0.11 ± 0.01 ^b^	2.30 ± 0.04 ^a^
C20:2n-6	Nd	0.02 ± 0.00 ^a^	Nd	0.02 ± 0.00 ^a^
C22:0	0.22 ± 0.02 ^a^	0.07 ± 0.00 ^b^	0.21 ± 0.03 ^a^	0.07 ± 0.02 ^b^
C20:3n-3	0.02 ± 0.00 ^a^	Nd	0.02 ± 0.00 ^a^	Nd
C20:4n-6	0.07 ± 0.00 ^a^	0.05 ± 0.00 ^c^	0.06 ± 0.00 ^b^	0.05 ± 0.00 ^c^
C24:0	0.08 ± 0.00 ^a^	0.03 ± 0.00 ^b^	0.08 ± 0.00 ^a^	0.04 ± 0.00 ^b^
∑ SFA	3.70 ± 0.10 ^a^	2.40 ± 0.01 ^b^	3.87 ± 0.12 ^a^	2.60 ± 0.17 ^b^
∑ MUFA	9.11 ± 0.12 ^c^	18.2 ± 0.16 ^a^	9.50 ± 0.11 ^c^	18.8 ± 0.20 ^b^
∑ PUFA	16.8 ± 0.12 ^b^	8.29 ± 0.11 ^d^	17.3 ± 0.13 ^a^	8.70 ± 0.10 ^c^
n-6/n-3	159 ± 11.9 ^a^	2.70 ± 0.04 ^c^	138 ± 6.77 ^b^	2.79 ± 0.10 ^c^
PUFA/SFA	4.51 ± 0.10 ^a^	3.41 ± 0.14 ^b^	4.45 ± 0.18 ^a^	3.31 ± 0.18 ^b^

F1 (soy protein isolate + sunflower oil); F2 (soy protein isolate + rapeseed oil); F3 (rapeseed protein isolate + sunflower oil); F4 (rapeseed protein isolate + rapeseed oil); SFA—saturated fatty acids; MUFA—monounsaturated fatty acids; PUFA—polyunsaturated fatty acids; n-6: omega-6 fatty acids; n-3: omega-3 fatty acids; Nd = not detected. ^a–d^ Mean values within the same row not followed by common letter differ significantly (*p* < 0.05).

**Table 6 foods-14-03841-t006:** TBARS value of chicken pâté samples during 60 days storage period.

Parameter	Treatments	Storage Time (Day)						
		0	30	45	60	T	F	T × F
TBARS (mg MDA·kg^−1^ sample)	F1	1.41 ± 0.03 ^ab^	1.48 ± 0.02 ^ab^	1.25 ± 0.14 ^abc^	1.13 ± 0.06 ^c^	*	*	*
	F2	1.37 ± 0.03 ^abc^	1.34 ± 0.21 ^abc^	1.22 ± 0.13 ^bc^	1.24 ± 0.17 ^abc^	*	*	*
	F3	1.40 ± 0.03 ^abc^	1.51 ± 0.02 ^a^	0.70 ± 0.04 ^d^	0.66 ± 0.07 ^d^	*	*	*
	F4	1.37 ± 0.14 ^abc^	1.46 ± 0.15 ^ab^	1.26 ± 0.14 ^abc^	0.75 ± 0.10 ^d^	*	*	*

F1 (soy protein isolate + sunflower oil); F2 (soy protein isolate + rapeseed oil); F3 (rapeseed protein isolate + sunflower oil); F4 (rapeseed protein isolate + rapeseed oil); TBARS—2-thiobarbituric acid reactive substances; T—time; F—formulation; T × F—interaction effect; ns = not significant; * *p* < 0.05. ^a–c^ Mean values within the same row not followed by common letter differ significantly (*p*< 0.05).

## Data Availability

The original contributions presented in this study are included in the article/Appendix A. Further inquiries can be directed to the corresponding author.

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
