# Peer review of "Rapeseed Protein Isolate as a Sustainable Alternative to Soy Protein: A Case Study on Chicken Pâtés"

_foods, 2025, doi:10.3390/foods14223841_

Round 1

Reviewer 1 Report

Comments and Suggestions for Authors

This study investigated the potential of rapeseed protein isolate (RPI) as a sustainable alternative to soy protein isolate (SPI) in chicken pâtés, considering the combined effects of protein (SPI or RPI) and oil (sunflower or rapeseed) sources. Research indicates that the proximate composition, pH, water activity (aw) and colour of the RPI-based formulations were largely comparable to their SPI counterparts and remained stable over 60 days of refrigerated storage. Furthermore, RPI-based formulations exhibited enhanced textural properties and improved oxidative stability, likely due to synergistic effects between rapeseed protein and oil. The substitution with rapeseed oil significantly improved the fatty acid profile, notably increasing α-linolenic acid (C18:3n3) and achieving a favourable n-6/n-3 ratio in F2 (2.7) and F4 (2.8). Sensory evaluation revealed that the formulation combining both rapeseed ingredients (F4) provided a stable and highly acceptable profile. These results collectively demonstrate that RPI is a viable and functional replacer for SPI, enabling the production of nutritious, high-quality, and sustainable chicken pâtés.

However, I therefore have to point out some comments:

Line 59-61: The description” Within this context, rapeseed (Brassica napus L.), the second most widely cultivated oilseed crop worldwide, has gained recognition as a promising source of plant-based proteins.” is too vague. The background description of RPI research lacks depth. Please supplement the key literature on RPI to demonstrate its advantages in this study and reinforce its necessity.

Line 61-62: The description” Protein isolates and concentrates derived from rapeseed have demonstrated considerable potential as meat extenders, contributing to improved nutritional value, desirable techno-functional characteristics, and greater sustainability in meat products.” is too simplistic. Please elaborate on the current state of research regarding rapeseed protein in meat products, particularly highlighting the research gaps in emulsified products, to underscore the necessity of this study.

Line 71-75: The mention of” Innovative processing methods, including enzymatic modification fermentation, and the removal of anti-nutritional factors prior to protein isolation, have been investigated to enhance both functional and sensory attributes of rapeseed proteins.” does not explicitly identify the current bottlenecks in the industrial application of RPI (such as the cost of removing antinutritional factors and the difficulty of flavor improvement), nor does it clearly explain how this research overcomes these challenges.

Line 76-77: The description” Chicken pâté, a traditional spreadable meat product, holds nutritional and sensory significance within the category of emulsified poultry foods.” lacks supporting documentation.

Line 113-115: The RPI production method only mentions “according to the procedure described by Đermanović et al., but produced on a larger scale in a pilot plant within the Instituteof Food Technology in Novi Sad,” without detailing adjustments to critical parameters during pilot testing (such as extraction pH, temperature, centrifugal speed, freeze-drying conditions, etc.). Please supplement the table of key process parameters for RPI preparation, specifying the extraction solvent, pH value, reaction time, purification steps, etc.

Line 118-120: The mention of “protein content ≥86% (dry basis)” does not specify the testing method, compromising experimental reproducibility. Please specify the specific method for detecting protein content.

Line 177-180: In the method for analyzing texture characteristics using a texture analyzer, only “The test parameters included a 5 kg load cell, a test temperature of 22℃, with a pre-test speed of 1.0 mm/s, test speed of 3.0 mm/s, and post-test speed of 10.0 mm/s. ” is mentioned, lacking supporting literature. Please supplement the literature for determining the parameters.

Line 221-222: The mention of” Panellists were trained on each selected descriptor prior to lexicon sorting. Lexicon sorting was performed using the Rate-All-That-Apply (RATA) method.” does not specify the training process for sensory evaluation panels (e.g., training duration, content, and consistency verification methods). Furthermore, the “RATA method screening attributes” section fails to list the specific names of the 25 attributes ultimately included in the descriptive analysis (DA) (e.g., only “rapeseed aroma” and “aftertaste intensity” are mentioned without providing a complete attribute list), making it impossible to assess the rationality of attribute selection.

Line 276-278:The description“Although differences among treatments were minor, higher ash values may be linked to differences in the mineral content of the raw materials used.” lacks supporting literature.

Line 283-285:The description“This pattern is consistent with typical proximate

composition analyses, where minor fluctuations in calculated carbohydrate content

largely result from changes in other major constituents.” lacks supporting literature.

Line 342-346:The description“Colour stabilization over time, particularly in……” lacks supporting literature.

Line 384-386: “However, during storage, a gradual decline in G′ was observed, particularly by day 60,……. assess the global viscoelastic response of the matrix under oscillatory stress.” fails to delve deeper into the underlying mechanisms by not incorporating microstructural factors such as protein aggregation states and fat globule distribution.

Part 3.4: In the section analyzing Textural and Rheological Behavior, it is recommended to integrate this with sensory analysis.

Line 410-411: The mention of” Given the critical role of oil fatty acid composition, particularly regarding cardio-vascular and metabolic outcomes in human health,” is too general. Please specify the specific categories of substances that affect biochemical processes and provide the necessary references. Please provide specific details on the effects of fatty acid composition in oils and fats on cardiovascular and metabolic health, along with supporting literature.

Line 457-458: The mention of” Lipid oxidation is a critical factor affecting the quality, shelf-life, and sensory attributes of meat products.” lacks of supporting literature. Please add relevant references to make the article more credible.

Line 469-474: The description” The significant reduction in TBARS after 45–60 days may

reflect MDA’s binding……decrease in measurable TBARS values.” is insufficient. Please provide statistical details, such as p-values or correlation coefficients, to support this conclusion.

Line 517-519: The description” While chemical composition often changes early during storage, sensory attributes, i.e. aroma, taste, and texture tend to exhibit more pronounced variations at later stages.” lacks supporting documentation.

Line 547: Consistency in using “Figure” or “Fig” for figure captions should be maintained.

Line 577-578: The description” despite having lower intrinsic protein content than SPI,

was able to deliver superior firmness and structural integrity in the pâtés.” lacks supporting data, merely stating that “both the RPI group and SPI group had an addition rate of 3% (w/w).” It fails to compare the effects of different RPI addition rates (e.g., 2%, 3%, 4%) on texture, making it impossible to verify the feasibility of “reducing the addition rate.”

Line 595-597: Add an explanation to the section” In addition, it highlights the potential for sustainable reformulation by replacing SPI with RPI,……and regulatory priorities.”

References

Line 617: The “In” preceding the journal title and “Elsevier” following it should be removed.

font size should be standardized and Use of bold type should be consistent.

The use of journal name abbreviations should be consistent. If " Trends Food Sci Technol" is abbreviated here, ensure all other journal names are treated similarly.

Comments on the Quality of English Language

no

Reviewer 2 Report

Comments and Suggestions for Authors

The manuscript presents a relevant and well organized study on the replacement of soy protein isolate (SPI) with rapeseed protein isolate (RPI) in chicken pâtés. The topic is timely, combining technological, nutritional, and sustainability perspectives that are of interest to the journal’s readership. The experimental design is solid, and the analytical work appears thorough and well executed.

Overall, the paper is scientifically sound and of potential interest for publication in Foods after minor to moderate revision. However, several clarifications and editorial improvements are required:

ABSTRACT

The abstract effectively summarizes the study; however, it would benefit from including one or two key numerical results and an explicit reference to the statistical significance of the main findings.

  1. Lines 13–14 (“... remained stable over 60 days of refrigerated storage, despite some initial differences.”)  The phrase is too vague. Please specify which differences were most relevant at day 0, for example, colour, pH, or TBARS and report one o more available data.
  2. Lines 17–18 (“... achieving a favourable n-6/n-3 ratio in F2 (2.7) and F4 (2.8).”) Clarify compared to which group (presumably F1). Also, indicate that the differences were statistically significant, specifying the probability level (p < 0.05, p < 0.01, or p < 0.001).
  3. Throughout the Abstract (references to F2, F4) The designations F2 and F4 appear without explanation. Since the abstract should be self-contained, either provide a brief clarification such as “F2 (soy protein isolate + rapeseed oil)” and “F4 (rapeseed protein isolate+ rapeseed oil)” upon first mention, or avoid using codes and describe the formulations directly (rapeseed oil compared to soy  oil ) .

INTRODUCTION

The Introduction is generally well structured and provides a coherent overview of the rationale for replacing soy protein isolate (SPI) with rapeseed protein isolate (RPI) in meat products. The logical flow from consumer demand and sustainability concerns to the functional and nutritional assessment of RPI in chicken pâtés is clear and scientifically valid.

However, for greater completeness, the authors should briefly address potential oxidative modifications that may occur during lipid and protein extraction from both soy and rapeseed.

MATERIALS AND METHODS

  1. Raw materials (around line 125) For completeness, the authors should consider (as supplementary table) the chemical composition of the raw materials used: amino acid profiles of SPI and RPI, fatty-acid composition of sunflower and rapeseed oils, and proximate composition of chicken meat and liver.
  2. Chemical analyses (Section 2.4) Because both plant and animal proteins are involved, specify the nitrogen-to-protein conversion factor used (e.g., 6.25 or other specific values).
  3. Colour measurements (Section 2.5) Indicate the illuminant (e.g., D65) used, and report calibration standards (white/black tiles) to ensure reproducibility.
  4. Fatty-acid analysis (Section 2.7) Include the internal standard used (e.g., C19:0 or C13:0). Clarify whether results are expressed as a percentage of total FAME or as grams of fatty acids per 100 g of sample (fresh weight).
  5. Statistical analysis (Section 2.9) Report the tests for normality and homogeneity of variance (Shapiro–Wilk, Levene). This is especially important for fatty-acid ratios (n-6/n-3) and sensory data.
  6.  To better interpret oxidation-related outcomes, please detail antioxidant additions during lipid extraction (e.g., BHT), headspace oxygen/light control during storage, and the time interval between jar opening and analysis.

RESULTS AND DISCUSSION

  1. Table 2 — Proximate composition

Harmonize superscript letters: in Moisture the smallest value carries “a”, whereas in Ash the largest value has “a”; in Protein the highest value lacks “a” and is marked “c”, while the next is “ab”. Likely, F1 (highest protein) should bear letter “a”.

  1. Tables 3, 4, 6 — Significance and formatting

Use lowercase letters for significance within the same row and uppercase letters for significance within the same column, applying the same rule in all tables to avoid confusion.

Differentiate significance levels (p < 0.05, p < 0.01, p < 0.001) using one, two, or three asterisks.

The column labeled “Interaction (T×F)” is unnecessary; remove it and state in the text that interactions were not significant.

For the “Formulation” significance column, clarify the meaning or remove it, since formulations are arranged in columns, not rows.

In Table 4, specify units explicitly: Firmness (N), Work of shear (N mm).

  1. Figure 1 — Colour differences

Apply the same significance notation as in Table 3.

ΔE* values should include standard deviations and significance indicators.

  1. Table 5 — Fatty-acid composition

The title reads “during 60 days storage period,” but only one value per formulation is presented. Specify whether values refer to day 0 or the mean across storage times.

The heading “Free fatty acids (%)” is incorrect: the data represent fatty acids in triglycerides, not free fatty acids, and the totals (~30 %) suggest expression as g/100 g product. Replace with “Fatty-acid composition (g per 100 g pâté, fresh weight)”.

If percentages of total FAME are intended, recalculate accordingly so that the sum equals 100 %.

Check internal consistency: the sum of individual PUFA does not perfectly match the total PUFA row.

Add definitions for n-6 and n-3 in the caption, and write fatty acids as “C18:2 n-6”, “C18:3 n-3”.

Remove the letter c after oleic and linoleic acid, since trans isomers are not reported.

Delete the “Other” category if no data follow.

Correct the caption typo “PFA” → “PUFA”.

  1. Figure 2 — Rheological parameters

Some histograms show letters from a to d where standard deviations seem small; re-check transcription errors.

If different significance indicators are used for time and formulation, clarify them in the figure legend to facilitate interpretation.

  1. Table 6 — TBARS values

Ensure the unit is reported correctly: mg MDA kg⁻¹ sample.

  1. Units and decimals (all tables)

Use consistent two-decimal precision across all numerical data.

All tables should be self-explanatory: include in each caption the meaning of F1–F4 and define all abbreviations.

DISCUSSION AND CONCLUSIONS

The discussion is generally sound and the conclusions are comprehensive. However, minor stylistic refinements are suggested to emphasize the novelty of applying RPI in emulsified poultry systems and to connect the results more directly to their technological and nutritional implications.

Quantitative results (e.g., n-6/n-3 = 2.8, p < 0.001) should be mentioned in the conclusion to strengthen the message.

REFERENCES

  1. Include the DOI for every reference (where available).
  2. Use journal abbreviations according to the Chemical Abstracts Service Source Index (CASSI).
  3.  Complete these one :
  1. Weiss, J. Emulsion Rheometry and Texture Analysis. University of Hohenheim. 2008.

28.Toldrá, F. Handbook of Meat Processing; John Wiley & Sons, 2010;

Round 2

Reviewer 1 Report

Comments and Suggestions for Authors

no

Comments on the Quality of English Language

no